# Genomic Analysis of *Aspergillus* Section *Terrei* Reveals a High Potential in Secondary Metabolite Production and Plant Biomass Degradation

**DOI:** 10.3390/jof10070507

**Published:** 2024-07-22

**Authors:** Sebastian Theobald, Tammi C. Vesth, Elena Geib, Jane L. Nybo, Jens C. Frisvad, Thomas O. Larsen, Alan Kuo, Kurt LaButti, Ellen K. Lyhne, Inge Kjærbølling, Line Ledsgaard, Kerrie Barry, Alicia Clum, Cindy Chen, Matt Nolan, Laura Sandor, Anna Lipzen, Stephen Mondo, Jasmyn Pangilinan, Asaf Salamov, Robert Riley, Ad Wiebenga, Astrid Müller, Roland S. Kun, Ana Carolina dos Santos Gomes, Bernard Henrissat, Jon K. Magnuson, Blake A. Simmons, Miia R. Mäkelä, Uffe H. Mortensen, Igor V. Grigoriev, Matthias Brock, Scott E. Baker, Ronald P. de Vries, Mikael R. Andersen

**Affiliations:** 1Department of Biotechnology and Bioengineering, Technical University of Denmark, 2800 Kongens Lyngby, Denmark; sebastiantheobaldsen@gmail.com (S.T.); tacv@novonesis.com (T.C.V.); jlnr@novonesis.com (J.L.N.); jcf@bio.dtu.dk (J.C.F.); tol@bio.dtu.dk (T.O.L.); kirstinelyhne@gmail.com (E.K.L.); igk@novonesis.com (I.K.); lineledsgaard@hotmail.com (L.L.); behen@dtu.dk (B.H.); um@bio.dtu.dk (U.H.M.); 2School of Life Sciences, University of Nottingham, Nottingham NG7 2RD, UK; elena.geib@nottingham.ac.uk (E.G.); matthias.brock@nottingham.ac.uk (M.B.); 3US Department of Energy Joint Genome Institute, Lawrence Berkeley National Laboratory, Berkeley, CA 94720, USA; akuo@lbl.gov (A.K.); klabutti@lbl.gov (K.L.); kwbarry@lbl.gov (K.B.); aclum@lbl.gov (A.C.); cindychen@lbl.gov (C.C.); mpnolan@lbl.gov (M.N.); lcsandor@lbl.gov (L.S.); alipzen@lbl.gov (A.L.); sjmondo@lbl.gov (S.M.); jlpangilinan@lbl.gov (J.P.); aasalamov@lbl.gov (A.S.); rwriley@lbl.gov (R.R.); ivgrigoriev@lbl.gov (I.V.G.); 4Fungal Physiology, Westerdijk Fungal Biodiversity Institute and Fungal Molecular Physiology, Utrecht University, 3584 Utrecht, The Netherlands; ad.wiebenga@gmail.com (A.W.); a.mueller@wi.knaw.nl (A.M.); roland.sandor.kun@gmail.com (R.S.K.); carolsantos218@hotmail.com (A.C.d.S.G.); 5Department of Biological Sciences, King Abdulaziz University, Jeddah 21589, Saudi Arabia; 6Environmental Molecular Sciences Division, Pacific Northwest National Laboratory, Richland, WA 99354, USA; jmagnuson@lbl.gov (J.K.M.); basimmons@lbl.gov (B.A.S.); 7US Department of Energy Joint Bioenergy Institute, 5885 Hollis St., Emeryville, CA 94608, USA; 8Lawrence Berkeley National Laboratory, Berkeley, CA 94720, USA; 9Department of Microbiology, University of Helsinki, Viikinkaari 9, 00014 Helsinki, Finland; miia.r.makela@helsinki.fi; 10Department of Plant and Microbial Biology, University of California Berkeley, Berkeley, CA 94720, USA

**Keywords:** genomics, fungi, *Aspergillus*, section *Terrei*, *Aspergillus terreus*, secondary metabolism, CAZymes

## Abstract

*Aspergillus terreus* has attracted interest due to its application in industrial biotechnology, particularly for the production of itaconic acid and bioactive secondary metabolites. As related species also seem to possess a prosperous secondary metabolism, they are of high interest for genome mining and exploitation. Here, we present draft genome sequences for six species from *Aspergillus* section *Terrei* and one species from *Aspergillus* section *Nidulantes*. Whole-genome phylogeny confirmed that section *Terrei* is monophyletic. Genome analyses identified between 70 and 108 key secondary metabolism genes in each of the genomes of section *Terrei*, the highest rate found in the genus *Aspergillus* so far. The respective enzymes fall into 167 distinct families with most of them corresponding to potentially unique compounds or compound families. Moreover, 53% of the families were only found in a single species, which supports the suitability of species from section *Terrei* for further genome mining. Intriguingly, this analysis, combined with heterologous gene expression and metabolite identification, suggested that species from section *Terrei* use a strategy for UV protection different to other species from the genus *Aspergillus*. Section *Terrei* contains a complete plant polysaccharide degrading potential and an even higher cellulolytic potential than other Aspergilli, possibly facilitating additional applications for these species in biotechnology.

## 1. Introduction

The fungal genus *Aspergillus* is of broad interest, as it not only contains species that are used in various industrial and pharmaceutical applications but also contains species capable of infecting humans and causing food and feed spoilage by mycotoxin production. Thereby, most *Aspergillus* species are assumed to synthesize more than 50 different secondary metabolites. An excellent example with all these traits is *Aspergillus terreus*. Primarily, *A. terreus* has attracted attention as an opportunistic human pathogen [1] but is also of interest through its biosynthesis of biotechnologically and pharmaceutically important products, in particular itaconic acid [2], itatartaric acid [3,4,5], and lovastatin/mevinolin, a secondary metabolite capable of lowering blood cholesterol levels [6]. Such statins are marketed and sold for the treatment of lifestyle diseases at an annual turnover of multibillion USDs [7]. Genome sequences already exist for several *A. terreus* strains [8,9,10,11], which has accelerated research on the species and sparked multiple genome-mining approaches to tap into the genomic wealth of *A. terreus* [12]. Analysis of the natural products produced by *A. terreus* has shown its capability of producing secondary metabolites with diverse activities, such as antiviral compounds [13,14], antitumor metabolites [15,16,17], and a wide range of mycotoxins, including patulin [18], citrinin [19], and geodin [20,21]. A. terreus also produces the phytotoxic, antifungal, and iron-reducing compound, terrein [22]. Due to the large quantities of terrein produced, regulatory elements from the *A. terreus* terrein biosynthesis gene cluster have been used to develop a heterologous expression system in *Aspergillus niger* suitable for the production and characterization of fungal secondary metabolites [22,23]. This expression system also supported the identification of a novel type of aspulvinone E-derived Asp-melanin, which protects the asexual conidia of *A. terreus* [24]. This Asp-melanin seems to discriminate *A. terreus* from other *Aspergillus* sections, in which a hydroxynaphthalene-melanin protects the conidia from UV and oxidative stress [25].

However, *A. terreus* is not the sole species member of this group, and closely related species are interesting candidates for genome analyses and mining. The *Aspergillus* section *Terrei* has been defined since 1985 and is characterized by columnar conidial heads in shades buff to brown [26,27], and at least 16 species (including *A. terreus*) have been assigned to this group [26,28].

Consistent with the name “*Terrei*”, nearly all species grouped into this section have been isolated from soil samples around the world [26], as well as from diverse sources like human wounds, wheat flour, sputum, a human ear, capybara droppings, corn, and the lung of a mouse [26]. However, while, among these species, *A. terreus* remained the only species with its genome sequenced, other species such as *A. alabamensis* and *A. allahabadii* are of particular interest. Both species have been identified as causes of invasive aspergillosis [29,30], and *A. alabamensis* has been identified as a producer of tremorgenic compounds [26]. Additionally, other species of the section *Terrei* show production of a large number and diversity of secondary metabolites [26]; thus, it is of interest to examine the genetic and chemical diversity in this section. In this study, we provide genome sequences of *A. alabamensis* [29], *A. allahabadii* [30], *A. ambiguus*, *A. aureoterreus*, *A. floccosus,* and *A. neoindicus* [26] and examine the potential for chemical diversity in these species, as defined by predicted gene clusters for secondary metabolite production, as well as its potential for plant biomass degradation. Furthermore, based on its genome sequence, a re-classification of a strain previously attributed to section *Terrei* was performed.

## 2. Materials and Methods

### 2.1. Fungal Strains

In this study, we sequenced *A. alabamensis* IBT 12702, *A. allahabadii* CBS 164.63, *A. ambiguus* CBS 117.58, *A. aureoterreus* CBS 503.65, *A. floccosus* CBS 116.37, and *A. neoindicus* CBS 444.75 as described by Samson et al. [26]. We further sequenced CBS 120.58 in the belief that it would be the species described as *A. microcysticus*, but sequencing revealed it to be a strain of *A. unguis*. We further compared the genome sequences to the published genomes, *A. clavatus* [11], *A. fumigatus Af293* [31] *A. terreus* [11], *A. flavus* [32], *A. ellipticus* [33], *A. neoniger* [33], *A. japonicus* [33], *A. nidulans* [34], *A. niger* ATCC 1015 [35], and *A. versicolor*. All sequences were downloaded from the Joint Genome Institute (JGI) MycoCosm portal [36].

### 2.2. Purification of DNA

For all sequences generated in this study, conidia stored at −80 °C were used to inoculate solid Czapek Yeast Agar (CYA) medium. Fresh conidia were harvested after 7–10 days and suspended in a 0.1% Tween solution. For the generation of biomass, a liquid CYA medium was inoculated and cultivated for 5–10 days at 30 °C. The mycelium was isolated by filtering through Miracloth (Millipore, 475855-1R, Merck, Søborg, Denmark), freeze dried, and stored at −80 °C until further use. DNA isolation was performed using a modified version of standard phenol extraction [37] and checked for quality and concentration using a NanoDrop (BioNordika, Herlev, Denmark).

### 2.3. DNA Sequencing and Assembly

All genomes in this study were sequenced with Illumina technologies. For all genomic Illumina libraries, 100 ng of DNA was sheared to 270 bp fragments using the Covaris LE220 (Covaris, Woburn, MA, USA) and size-selected using SPRI beads (Beckman Coulter, Brea, CA, USA). The fragments were treated with end-repair, A- tailing and ligated to Illumina compatible adapters (IDT, Inc., San Diego, CA, USA) using the KAPA-Illumina library creation kit (KAPA biosystems, Woburn, MA, USA). The prepared libraries were quantified using KAPA Biosystem’s next-generation sequencing library qPCR kit and run on a Roche LightCycler 480 real-time PCR instrument. The quantified libraries were then multiplexed with other libraries, and the library pools were prepared for sequencing on the Illumina HiSeq sequencing platform utilizing a TruSeq paired-end cluster kit, v3, and Illumina’s cBot instrument to generate clustered flow cells for sequencing. Sequencing of the flow cells was performed on the Illumina HiSeq2000 sequencer using a TruSeq SBS sequencing kit, v3 (Illumina, San Diego, CA, USA), following a 2 × 150 indexed run recipe. After sequencing, the genomic fastq files were quality-control-filtered to remove artefacts/process contamination and assembled using Velvet [38]. The resulting assemblies were used to create in silico long mate-pair libraries with inserts of 3000 ± 90 bp that were then assembled with the target fastq using AllPathsLG release version R47710 [39]. All genomes were annotated using the JGI annotation Pipeline [36]. The genome assembly and annotations are available at the JGI fungal genome portal MycoCosm [36].

### 2.4. Whole-Genome Phylogeny

The protein sequences of all organisms were compared using BLASTp (e-value cut-off 1 × 10^−5^). Orthologous groups of sequences were constructed based on best bidirectional hits (BBH). Two hundred groups with a member from each species were selected and the sequences of each organism were concatenated into one long protein sequence. The concatenated sequences were aligned using MAFFT v7 (thread 16), and well-aligned regions were extracted using Gblocks (-t = p -b4 = 5 -b5 = h). Trees were then constructed using multi-threaded RAxML v8, the PROTGAMMAWAG model, and 100 bootstrap replicates.

### 2.5. Prediction of Secondary Metabolite Biosynthesis Gene Clusters

For the prediction of secondary metabolite (SM) biosynthesis gene clusters, we developed a command-line Python script roughly following the SMURF algorithm [40]. As input, the program takes genomic coordinates and the annotated PFAM domains of the predicted genes. Based on the multi-domain PFAM composition of identified ‘backbone’ genes, it can predict seven types of SM clusters: (1) polyketide synthases (PKSs), (2) PKS-like, (3) nonribosomal peptide-synthetases (NRPSs) (4) NRPS-like, (5) hybrid PKS–NRPS, (6) prenyltransferases (DMATS), and (7) terpene cyclases (TCs). Besides backbone genes, PFAM domains, which are enriched in experimentally identified SM clusters (SM-specific PFAMs) were used in determining the borders of gene clusters. The maximum allowed size of intergenic regions in a cluster was set to 3 kb, and each predicted cluster was allowed to have up to 6 genes without SM-specific domains.

### 2.6. Generation of SMGC Families

The proteins of the resulting secondary metabolism gene clusters (SMGCs) were compared among each other by alignment using BLASTp (BLAST+ suite version 2.2.27, e-value ≤ 1^–10^). Subsequently, a score based on BLASTp identity and shared proteins was created to determine the similarity between gene clusters: To create a cluster similarity score, a combined score of tailoring and backbone enzymes was created. The sum of the BLASTp percent identity (pident) of all hits for tailoring enzymes between two clusters was divided by the maximum number of tailoring enzymes (nmax) and multiplied by 0.3 (Equation (1)).
(1)$$score\;(tailoring)=\frac{sum(pident_{tailoring})}{n{max}_{tailoring}}*0.3\;$$

Then, the score for the backbone enzymes was calculated in the same manner but multiplied by 0.7 to give more weight to the backbone enzymes (Equation (2)).
(2)$$score\;(backbone)=\frac{sum(pident_{backbones})}{nmax_{backbones}}*\;0.7$$

The scores were added to create an overall cluster similarity score (Equation (3)).
(3)$$Similarity\;score=\frac{sum(pident_{tailoring})}{n{max}_{tailoring}}*0.3+\frac{sum(pident_{backbones})}{nmax_{backbones}}*\;0.7$$

Using these scores, we created a weighted network of SMGC clusters and used a random walk community detection algorithm (R version 3.3.2, igraph_1.0.1) [41] to determine the families of the SMGCs. Finally, we ran another round of random walk clustering on the communities which contained more members than species in the analysis. YWA and emodin gene clusters from MIBiG were identified inside the families using BLASTP, and their members were labelled as gene clusters producing similar compounds.

### 2.7. Genetic Dereplication

The MIBiG [40] genbank files were downloaded, parsed using Biopython V1.76 [42], and the SMGC entries of *Aspergillus* and *Penicillium* species were searched against a database of all SMURF annotated proteins of the dataset using BLASTp [43]. BLAST hits with over 95% identity were annotated with their respective MIBiG query sequence. All SMGCs inside a family of the hit were annotated as related to the MIBiG query.

### 2.8. Heterologous Expression of Pigment Biosynthesis Genes

To heterologously express *melA* or *pksP* homologous genes in *A. niger* ATNT16, the respective genes were amplified with Phusion high-fidelity DNA polymerase (Thermofisher Scientific, Waltham, MA, USA) from genomic DNA. Oligonucleotides were designed containing 15 bp overhangs to the *Nco*I site of the SM-Xpress_URA plasmid to allow site-specific assembly via in vitro recombination using the In-Fusion HD cloning kit (Takara/Clonetech, San Jose, CA, USA). Plasmids were propagated in *Escherichia coli* DH5α and isolated using the NucleoSpin Plasmid kit (Macherey-Nagel, Allentown, PA, USA). Circular plasmids were used for the transformation of *A. niger* ATNT16 Δ*pyrG,* following the transformation protocol as described previously [23]. Regeneration and selection of positive transformants were achieved on an *Aspergillus* minimal medium (AMM) agar with osmotic stabilization by 1.2 M sorbitol and without uridine supplementation. Transformants were checked for integration of expression constructs by diagnostic PCR. An *A. terreus melA*-expressing ATNT16 transformant was generated in a previous study [23].

### 2.9. Promoter Exchange in A. allahabadii

To express the *A. allahabadii pksP* homologue under the control of the *A. fumigatus pksP* promoter, two PCR products were generated using the Phusion high-fidelity DNA polymerase (Thermo Fisher Scientific, Loughborough, UK): 1000 bp promoter sequence from *pksP* (P*pksP*) from the genomic DNA of *A. fumigatus* Af293 and *pksP* from *A. allahabadii* including the *A. nidulans trpC* terminator from the heterologous expression plasmid. The PCR products were mixed with a *Hind*III-digested *ptrA*_pUC19 plasmid [44] and assembled by the In-Fusion HD-cloning kit (Takara/Clonetech). Plasmid amplification in *E. coli* and fungal transformation were carried out as described for the heterologous expression of pigment biosynthesis genes in *A. niger* except that 0.1 µg/mL pyrithiamine was used as the selection marker. The transformants were streaked three times on the pyrithiamine-containing AMM agar and checked for construct integration with oligonucleotides 30 + 31 using Phire Green Hot Start II DNA polymerase (Thermo Fisher Scientific).

### 2.10. Chromatographic Analysis of Transformants

Selected transformants were grown for 26 h to 46 h in the AMM(-N) medium containing 50 mM of glucose, 10 mM of glutamine, and 10µg/mL of doxycycline [23]. The mycelium was removed by filtration over Miracloth filter gauze (Merck), and culture filtrates were extracted with an equal volume of ethyl acetate. The organic phase was collected and dried over sodium sulphate. After evaporation of the solvent under reduced pressure, the residue was solved in 1 mL methanol. An HPLC analysis was carried out as described previously using a Dionex UltiMate3000 (ThermoFisher Scientific, Waltham, MA, USA) and an Eclipse XDB-C18 column, 5 μm, 4.6 × 150 mm (Agilent, Glostrup, Denmark) that was kept at 40 °C [23].

### 2.11. Prediction of Encoded CAZymes

CAZymes were predicted for all genomes using the Carbohydrate-Active Enzymes (CAZy) database [45,46] and the method described in [33]. Growth profiles were performed on minimal medium [47] with 25 mM of the mono- and oligosaccharides or 1% of the polysaccharides and crude substrates.

### 2.12. Growth Profiling on Plant Biomass Related Compounds

The growth of the species was analyzed using the *Aspergillus* minimal medium [47], with the carbon sources as indicated in the figure. Monosaccharides and disaccharides were added at 25 mM final concentration, while polysaccharides and crude substrates were added as 1% (*w*/*v*) final concentration.

## 3. Results and Discussion

### 3.1. Genome Sequencing and Genome Statistics

We prepared genomic DNA from seven fungal species that were assumed to belong to section *Terrei*: *A. alabamensis*, *A. allahabadii*, *A. ambiguus*, *A. aureoterreus*, *A. floccosus*, *A. microcysticus* CBS 120.58 (later re-annotated as *A. unguis*), and *A. neoindicus*. All genomes were Illumina-sequenced, assembled, and annotated using the JGI fungal genome pipeline [36,48]. All genomes are available on NCBI, the JGI genome portals, and MycoCosm [36]. Full statistics of the sequenced genomes and further genomes used for the comparative analysis are available in Appendix A. An overview on the most important statistics across 18 fungal species is provided in Figure 1.

Unexpectedly, *A. microcysticus* CBS 120.58 had previously been described as a member of the section *Terrei* [26], but closer examination of the genome revealed that it represents *A. unguis* of the section *Nidulantes*. Therefore, it was kept as a reference, but its genome was not included in the comparisons of the six new *Terrei* genomes. Overall, the quality-related genome statistics (Figure 1, Appendix A) revealed a high quality of the new genome drafts. Scaffold numbers are between 43 and 224, and scaffold N50 was generally between 11 and 16, with the exception of *A. floccosus* with N50 = 39. This indicates an excellent assembly of the genomes, which is comparable to or even better than many other published fungal genomes. When compared to genomes from *A. flavus* and *A. clavatus,* there is a similar number of scaffolds (Figure 1) and average scaffold length, but the median scaffold length is 50–100 times higher for these new genomes (Appendix A). It was furthermore possible to assign InterPro predictions of function to >75% of predicted proteins, which is similar or better than possible for previously sequenced *Aspergillus* genomes.

When examining the content and characteristics of the genomes of species from section *Terrei*, the genomes appear quite homogenous. The genome sizes are around 30 Mb, whereas the genome size from *A. alabamensis* is slightly expanded, with 32 Mb. All species have a similar number of predicted proteins (in a range of 761 proteins), while the number for *A. terreus* is slightly lower with 10406 proteins (Figure 1). However, the genome of *A. terreus* was sequenced and annotated by using different bioinformatic pipelines, as applied in this study. All genes from the species of section *Terrei* also have a common average number of 3.2 exons per gene. In summary, these genome analyses indicate that the section *Terrei* comprises a very homogeneous section, particularly when compared to section *Nigri*, for which we previously described a substantially larger diversity [33].

### 3.2. Species Phylogeny

Based on the predicted proteins, we were interested in examining the species diversity and examining the phylogenetic relationship of the species at the genome level. Particularly, given that the genome statistics are similar for the new genomes, it was interesting to see whether the proposed species are more diverse at the sequence level. To achieve this, we built a whole-genome-based phylogeny of a total of 18 species across the Aspergilli (Figure 1A) that was based on 200 genes, from which exactly one ortholog was found in each species (also known as monocore genes). The resulting tree is in accordance with a calmodolin-based tree presented in a previous study [26] and the phylogenetic analysis of some section *Terrei* isolates by Varga et al. [52]. It additionally supports the proposition that *A. ambiguus*, *A. aureoterreus*, *A. floccosus*, and *A. neoindicus* should indeed be recognized as separate species, as there is a clear separation of these species in the phylogram. This analysis further revealed that, despite having genome statistics similar to the other section *Terrei* species from this study, *A. ambiguus* is the most distant species within this group.

### 3.3. Species in Section Terrei Are Rich in Genes Encoding Secondary Metabolite Clusters

Analyses of secondary metabolism of *A. terreus* already showed a high diversity in the products formed. Therefore, we were interested in mapping genes responsible for the diversity of secondary metabolism in section *Terrei* relative to *A. terreus* and other sections of Aspergilli. Secondary metabolite gene clusters (SMGCs) were predicted based on a SMURF-like prediction method [38]. In total, we predicted 3275 genes in the six new genomes potentially involved in secondary metabolism (Appendix A). So-called backbone genes, including synthases, synthetases, and cyclases, that are involved in producing the backbone compound to be modified by other cluster encoded enzymes were counted and sorted into categories based on predicted functions (Figure 2A). In total, we found 495 backbone genes in the new genomes, which indicates that the species of section *Terrei* are similarly rich in secondary metabolism as the secondary metabolism-rich section *Nigri* (Figure 2A). In particular, *A. floccosus* revealed 108 backbone genes across 88 predicted clusters. As far as we are aware, this is the highest number of genes found for secondary metabolite production in a single *Aspergillus* species. Intriguingly, this enriched number comes from an expansion of nearly all classes of secondary metabolite backbones, whereby the genome itself is not particularly expanded as seems to be the case for genomes of species from section *Nigri*. Since *A. floccosus* has not been reported to produce a large number of known secondary metabolites [26], our analyses suggest that several of these genes and gene clusters may be responsible for the production of new compounds.

We further analyzed how many of the gene clusters are shared among species—either within section *Terrei* or with other Aspergilli. All predicted clusters in the set (1058) were compared to each other, and we calculated how many are shared in either direction in all two-way comparisons (Figure 2B). In this comparison, it became apparent that only 50–80% of gene clusters are shared between any two compared species in section *Terrei*, showing a unique SM production potential in all of the *Terrei* species included here. However, the internal similarity in section *Terrei* is more homogeneous than that within the four members of section *Nigri*, which only share 35–60% of their SM capacity. Furthermore, only 10–30% of the SMGCs are shared with the other *Aspergillus* species included as references (Figure 2B).

### 3.4. Secondary Metabolite Clusters in New Terrei Genomes Can Be Sorted into 167 Families

As the genomes of species from section *Terrei* appear to be rich in secondary metabolite genes, we wanted to see how much of the diversity was replicated between species. In a first step, it was determined which of the gene clusters across genomes are variations of each other. Based on a previously developed method [53], gene clusters that possess homologous genes for backbone enzymes and contain homologous genes for tailoring enzymes were identified. This strategy followed the hypothesis that highly similar clusters are expected to produce related compounds. Including this assumption provides a more accurate measure of the chemical diversity of the isolates. In total, the 495 SM biosynthetic genes in the seven section *Terrei* genomes can be collapsed into 167 families (Appendix A), thus suggesting that at least 167 different chemical moieties can be produced by the seven species. Of these families, 88 are only found in a single genome, with each new genome containing 9–16 SMGCs that are not found in any of the other genomes in the comparative analysis (Appendix A). The 53% (88/167) of clusters only found in single genomes indicate a potential for high chemical diversity and unique chemistry in the section *Terrei*.

### 3.5. Genetic Dereplication of the Terrei SM Gene Clusters by Comparison with the MIBiG Database

The analysis of the SMGC families above led to the hypothesis that the SMGCs of section *Terrei* primarily contain unknown clusters, which was specifically supported by the finding of many unique clusters and only a few clusters shared between many species (Appendix A). We thus performed what can be called „genetic dereplication” of the gene clusters [53], in which the families of SMGC from section *Terrei* were compared to the characterized gene clusters available in the MIBiG database [54]. We dereplicated all clusters across the full dataset of 18 species and extracted the information regarding cluster families from section *Terrei* (Figure 3, see Appendix A for details on the association of all clusters to compounds in the complete dataset).

### 3.6. Pathogenicity Factors from Terrei

A particularly important metabolite in *Terrei* is acetylaranotin. While gliotoxin, produced by several pathogenic strains of section *Fumigati*, is important for the infection of human lungs [55], it seems that the analogous secondary metabolite acetylaranotin (also with a di-sulphur bridge) may be important for the infection of human lungs (in immunocompromised patients) by species in section *Terrei* (Figure 3). These two types of di-sulphur bridge antioxidant molecules (acetylaranotin and gliotoxin) have similar gene clusters ([56] and this study) but are section-specific for *Terrei* and *Fumigati*, respectively.

This analysis confirmed our hypothesis that section *Terrei* contains a high potential for unique natural product chemistry; out of the 167 identified SMGC families, only 17 (10%) could be connected to characterized clusters (Figure 3, Appendix A). Some interesting observations were made during this analysis: Terreic acid, patulin, and 6-methylsalicylic acid (6-MSA) are found in the same family. This supports the composition of SMGC families, as terreic acid and patulin are known to be produced from 6-MSA [56,57,58] Furthermore, all *Terrei* species have an SMGC that is homologous to the terrequinone cluster described from *A. nidulans* [59]. A previous study [26] identified terrequinones in several species from section *Terrei* with the exception of *A. floccosus*, *A. allahabadii,* and *A. neoindicus*. The genome analysis, however, suggests that these species may also be capable of producing terrequinone derivatives, but production seems to be induced under yet unknown conditions. Finally, we were interested in the formation of melanins protecting the *Aspergillus* conidia from different environmental stressors. While a naphthopyrone-derived dihydroxynaphthalene (DHN-) melanin appears common to species from various *Aspergillus* sections [60,61,62], previous studies revealed that *A. terreus* produces a novel type of melanin. This Asp-melanin derives from aspulvinone E which is produced by the NRPS-like enzyme MelA and subsequently polymerised by a tyrosinase [24]. In contrast, no homologue for a naphthopyrone synthase such as WA from *A. nidulans* or PksP from *A. fumigatus* was detected in *A. terreus* [24,63]. Therefore, we had a closer look at PksP and MelA homologues in species from section *Terrei*.

### 3.7. Pigments in Conidia from Section Terrei

To elucidate whether Asp-melanin is specific only for *A. terreus* or common to the section *Terrei*, we searched genomes for homologues of the *A. terreus* aspulvinone E synthetase MelA as well as for the naphthopyrone synthase PksP from *Aspergillus fumigatus*. The genomes of *A. alabamensis*, *A. aureoterreus*, *A. floccosus,* and *A. ambiguus* indeed contained a MelA homologue (Figure 3), while a PksP homologue was additionally detected in the *A. ambiguus* genome (Appendix A). *A. allahabadii* exclusively revealed a PksP homologue. Neither of both genes was detected in *A. neoindicus*. In agreement, species containing the *melA* homologue mostly produced yellow to brown pigmented conidia (Figure 4A), whereas *A. unguis* (as belonging to section *Nidulantes* and carrying a PksP homologue) produced green and *A. neoindicus* white conidia. Unexpectedly, despite the presence of a *pksP* homologue in *A. allahabadii*, the conidia also appeared non-melanized. To verify our pigment biosynthesis predictions from in silico analyses, the respective *melA* and *pksP* homologues were amplified from genomic DNA and cloned into the *Aspergillus niger* expression platform strain ATNT16 for heterologous metabolite biosynthesis [23,64]. Metabolite analyses confirmed that all *melA* homologues produced aspulvinone E and its UV-interconvertible *cis*-isomer, isoaspulvinone E (Figure 4B). In addition, by taking the PksP product from *A. fumigatus* as reference, all *pksP* homologues revealed the production of YWA1 (Figure 4C). However, all transformants producing the PksP homologue from *A. ambiguus* revealed low product yields suggesting a comparably low activity of this enzyme. As *A. ambiguus* also contains a *melA* homologue, and the conidia appear brownish, the production of Asp-melanin may be dominating in this fungus.

Unexpectedly, while the conidia from *A. allahabadii* appeared unpigmented, its PksP homologue produced YWA1 (Figure 4C). Therefore, it appeared likely that the *pksP* gene is not expressed in *A. allahabadii* under conidia-forming conditions. To confirm this assumption, we aimed for an induced expression of the *pksP* homologue in *A. allahabadii* by replacing its native promoter with the *pksP* promoter from *A. fumigatus*. Indeed, in agreement with a yellow colour of compound YWA1, *A. allahabadii* transformants produced yellow conidia with some green sectors (Figure 4D). This indicates that the native *pksP* promoter from *A. allahabadii* is not induced during conidiation, but downstream genes required for the formation of DHN-melanin seem to be active.

Interestingly, the production of different melanin pigments follows the phylogenetic tree shown in Figure 1. *A. ambiguus*, the species most distantly related to *A. terreus,* seems to produce DHN-melanin and Asp-melanin in parallel. In contrast, *A. allahabadii* and *A. neoindicus* produce white conidia, despite the presence of a DHN-melanin biosynthesis pathway in *A. allahabadii*. All other species from section *Terrei* lack a PksP homologue but gained or evolved a MelA protein. This indicates a complete loss of the DHN-melanin biosynthesis pathway in core species from section *Terrei*. As DHN-melanin inhibits acidification of phagolysosomes in macrophages and amoeba [24,63], these properties appear unfavourable for species from section *Terrei*. However, as a complete loss of conidia pigmentation causes increases sensitivity against oxidative and UV-stress, Asp-melanin has evolved as alternative protective mechanism. Therefore, these analyses indicate that Asp-melanin is a feature that can be specifically attributed to species from this section.

### 3.8. Aspergillus Section Terrei Has a High Plant Biomass Polysaccharide-Degrading Potential

In contrast with species from sections *Nigri* [33] and *Flavi* [65], species from section *Terrei* have not been as extensively studied for the conversion of plant-biomass-related polysaccharides. Most of these studies are related to *A. terreus,* focusing on a wide range of topics, such as the production and characterization of the enzymes of this species acting on various polysaccharides [66,67,68,69] and the degradation of lignocellulose [70,71,72]. The presence of the carbohydrate-active enzymes (CAZymes)-encoding genes in the genomes of the six *Terrei* species was analyzed and compared to 10 other Aspergilli to determine the variation of CAZymes within and outside section *Terrei*. The CAZy content was compared to the growth profiles of these strains on plant-biomass-related substrates. Overall, all species of section *Terrei* have a comparable CAZy content, especially with respect to glycoside hydrolases, glycosyl transferases, polysaccharide lyases, and carbohydrate esterases, with more variations in the number of carbohydrate binding modules and auxiliary activities (Appendix A(1)). The numbers of these candidate enzymes are more similar in section Terrei than was observed for section *Nigri* [33], *Flavi* [65], *Usti,* and *Nidulantes* (unpublished data). 

To evaluate the differences in the hydrolytic potential of the different species for plant biomass degradation in more detail, the CAZy families that are specific for certain polysaccharides (Appendix A(8–14)) were used to establish a polysaccharide-degradation profile for each of the species (Figure 5, Appendix A(15)). This confirmed the high similarity in section *Terrei*, with the exception of its most distant species, *A. ambiguus*, which mainly had a reduced number of putative cellulases, while the other species had a higher number of cellulases compared to other Aspergilli. Also, the total number of hemicellulases in these species is higher than in other tested Aspergilli, with the exception of *A. versicolor* (Figure 5). These data suggest that section *Terrei* may also have significant potential for plant biomass conversion in biotechnology, as these species contain a complete set of plant-biomass-degrading enzymes covering all plant polysaccharides.

The general ability of species from section *Terrei* to convert plant biomass is also reflected in their growth profile, showing overall good growth on all tested plant-biomass-related substrates, although growth was better in general for A. *allahabadi* and *A. ambiguus* (Appendix A). Growth was observed on nearly all monosaccharides, suggesting a complete sugar catabolism, similar to what has been described for *A. niger* and *A. nidulans* [73]. The inability of many fungi to grow on D-galactose from spores has been commonly observed, and it was shown in *A. niger* that this is due to the lack of D-galactose uptake during germination [74]. Interestingly, *A. ambiguus* shows good growth on D-galactose, while the other species show no to poor growth. As this study demonstrated that *A. ambiguus* is the most distant species within section *Terrei*, this may suggest an evolutionary adaptation in *A. ambiguus*. All strains show very good growth on crude plant biomass, supporting their potential as enzyme cocktail producers for plant biomass valorization. In contrast, only *A. aureoterreus* shows good growth on inulin, despite having a similar number of inulinases as several other species. This indicates that expression of the genes may have a larger role on this phenotype than the genome content. A transcriptome-profiling study of *A. terreus* showed clear differential expression of CAZyme-encoding genes on different plant biomass substrates [75], indicating the relevance of the regulation of gene expression in their plant biomass degradation approach. This was also observed at the enzyme production level in an exoproteomic analysis of 20 Aspergilli, including *A. terreus*, which revealed that very few orthologous CAZymes are produced under the same conditions in different species and even in different strains of the same species [76].

## 4. Conclusions

Genome analyses of *Aspergillus* section *Terrei* confirmed that these species are very rich in secondary metabolism genes. We established 167 distinct families, with most of them corresponding to potentially unique compounds or compound families. It also revealed a different UV protection approach in section *Terrei* compared to other Aspergilli, pointing to the parallel evolution of a survival trait and the existence of multiple protection strategies, which get exchanged between microbial species through horizontal transfer, followed by a selection of the one that suits the ecological niche of the host the best.

Section *Terrei* further contains a complete plant-biomass-degrading potential and an even higher cellulolytic potential than other Aspergilli. It is clear that the higher number of CAZymes are stemming from an expansion within all polysaccharides we examined, possibly facilitating additional applications for these species in biotechnology.

In summary, section *Terrei* species have a more similar genome content within the section than species in other sections of *Aspergillus* but contain a highly interesting biotechnological potential. We see that this work may support further studies into this section of fungi and examine the abundant diversity of unique secondary metabolite gene clusters, CAZymes, and evolutionarily distinct strategies.

## Figures and Tables

**Figure 1 jof-10-00507-f001:**
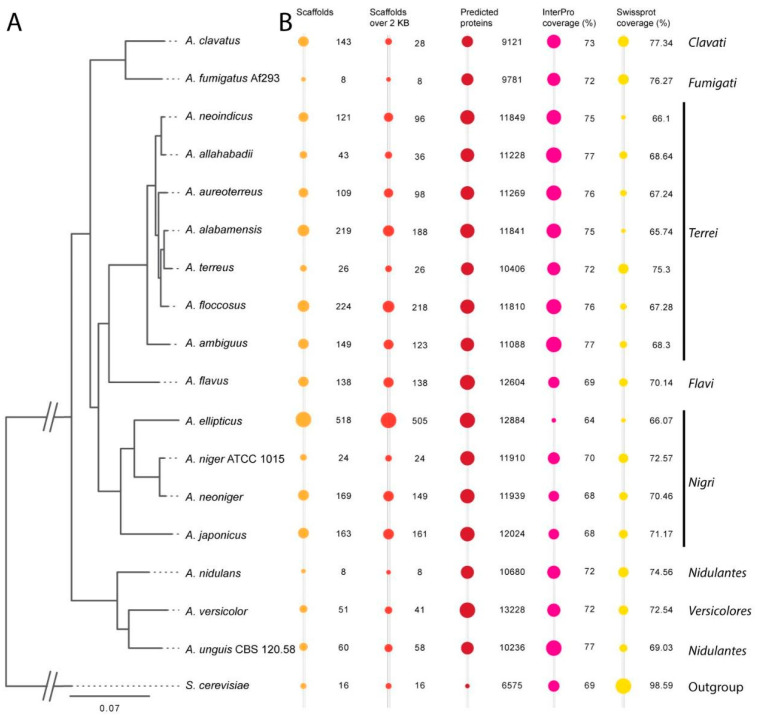
Dendrogram and bubble plots illustrating phylogenetic distances among seven genomes from section *Terrei*. Ten other Aspergilli representing different sections are added to place the *Terrei* in the genus-wide phylogeny. *S. cerevisiae* is added as an outgroup. Additional information is available in Appendix A. (**A**)—Phylogenetic tree created using RAxML [49], MAFFT [50], and Gblocks [51] based on 200 conserved genes found in one copy in each of the genomes (monocore genes). (**B**)—Bubble plots of key genome statistics. The bubble sizes are scaled within the categories and are not comparable across categories.

**Figure 2 jof-10-00507-f002:**
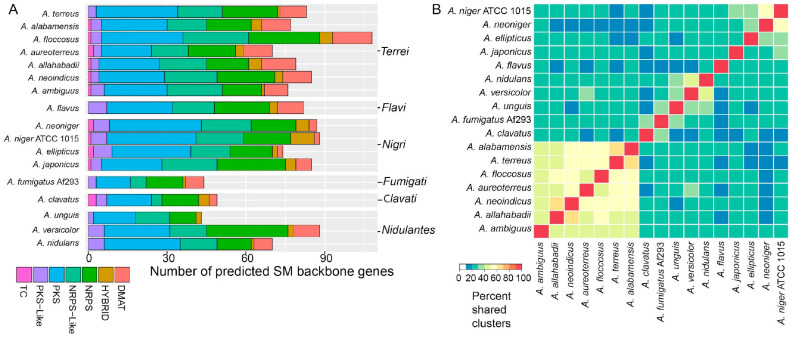
Summary of secondary metabolism-related genes and gene clusters in section *Terrei* and reference species. (**A**) Total secondary metabolite (SM) gene cluster family profile of Aspergilli classified by backbone enzyme type. (**B**) The heatmap is clustered horizontally and vertically according to the percentage of shared gene clusters. Note that the clustering generally follows the phylogeny of Figure 1. DMAT: Dimethylallyltransferase (Prenyl transferases); HYBRID: Gene-containing domains from NRPS and PKS backbones in any order; NRPS: Non-ribosomal peptide synthetase; NRPS-like: Non-ribosomal peptide synthetase like containing an adenylation and condensation domain and either a C-terminal thioesterase or reductase domain; PKS: Polyketide synthase; PKS-like: Polyketide synthase like, containing at least two PKS specific domains and another domain; TC: Terpene cyclase.

**Figure 3 jof-10-00507-f003:**
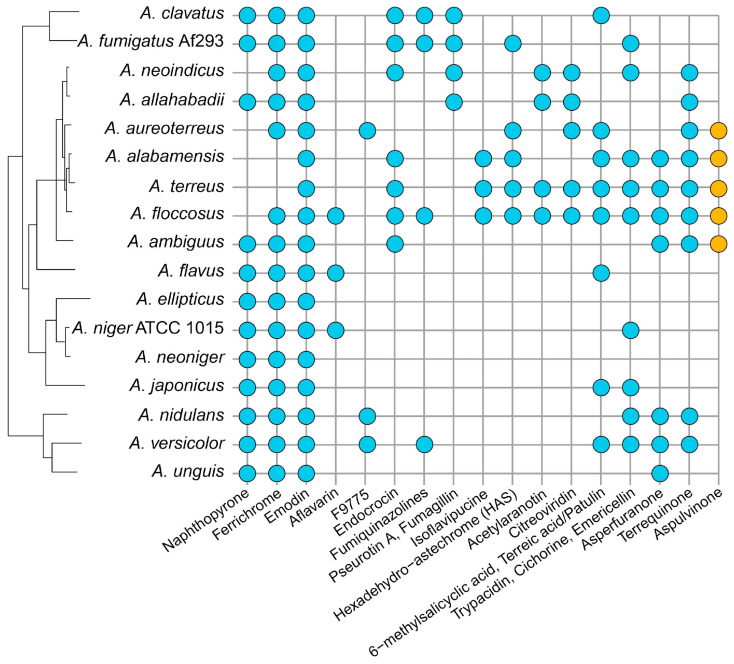
Association of gene clusters from section *Terrei* to previously assigned and characterized orthologous gene clusters. All gene families were compared to the MIBiG database to identify clusters homologous to characterized gene clusters. Gene cluster families with an associated characterized cluster are marked with a dot for all species containing a member of the gene cluster family, and, thus, presumably also a proposed compound. The family associated with Asp-melanin production from aspulvinone E (orange) was added manually through searching for the *A. terreus* aspulvinone E synthetase MelA. The secondary metabolites mentioned in Figure 3 are not necessarily exactly the same; for example, aflavarins are produced by species section *Flavi* (in the sclerotia), while species in section *Nigri* produce the very similar kotanins in the mycelium (but the gene clusters for those polyketides are quite similar).

**Figure 4 jof-10-00507-f004:**
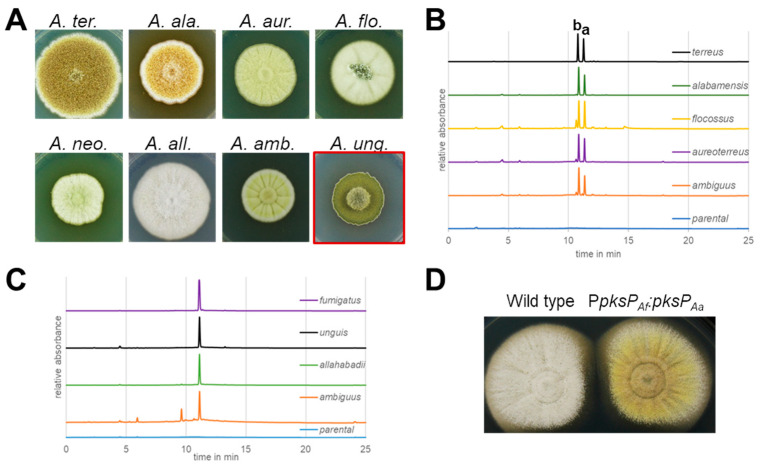
Analysis of conidia pigmentation in section *Terrei*. (**A**) Phenotype of colonies of species from section *Terrei* and of *A. unguis* from section *Nidulantes* on complete *Aspergillus* medium. *A. ter*. = *A. terreus*, *A. ala*. = *A. alabamensis*, *A. aur*. = *A. aureoterreus*, *A. flo*. = *A. floccosus*, *A. neo*. = *A. neoindicus*, *A. all*. = *A. allahabadii*, *A. amb*. = *A. ambiguus*, *A. ung*. = *A. unguis*. (**B**) HPLC analysis of products formed from the heterologous expression of *melA* homologues from section *Terrei* in the *A. niger* expression platform ATNT16 (parental). a—aspulvinone E, b—isoaspulvinone E. (**C**) HPLC analysis of products formed from the heterologous expression of pksP homologues from section *Terrei* and from *A. unguis* and *A. fumigatus* in *A. niger* ATNT16 (parental). (**D**) *A. allahabadii* wild type and transformant with a fusion of the *A. fumigatus pksP* promoter with the *A. allahabadii pksP* gene (P*pksP_Af_*:*pksP_Aa_*).

**Figure 5 jof-10-00507-f005:**
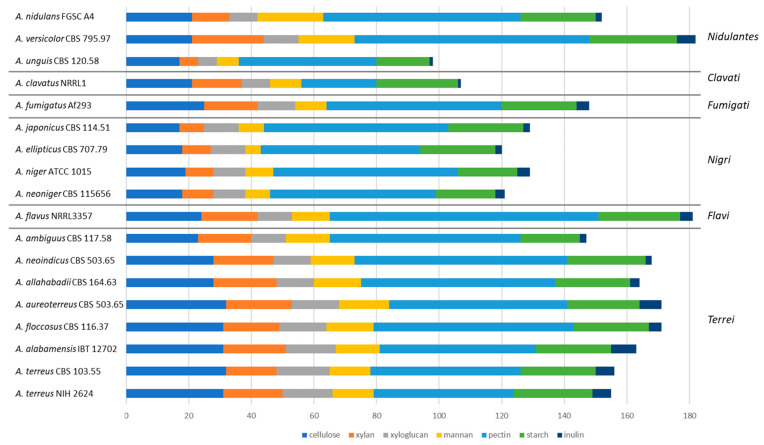
Comparison of plant polysaccharide degrading potential of section *Terrei* and reference species. The x-axis presents the number of genes in polysaccharide-specific CAZy families. The horizontal lines separate the sections within the genus *Aspergillus*.

## Data Availability

All new genomes of this study are available through the JGI MycoCosm portal [36]: https://mycocosm.jgi.doe.gov/mycocosm/home (accessed on 14 January 2024).

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
