# Peer review of "Genomic Analysis of Aspergillus Section Terrei Reveals a High Potential in Secondary Metabolite Production and Plant Biomass Degradation"

_jof, 2024, doi:10.3390/jof10070507_

Round 1

Reviewer 1 Report

In this research, the authors present draft genome sequences for six species from Aspergillus section Terrei and one species from Aspergillus section Nidulantes. The work looks quite interesting, but in its current form there are a lot of questions about it. Therefore, it should not be accepted for publication without serious revision

1. Discussion is too short. This part should be improved and contained the comparison with the similar articles.

2. Are there any broader implications or applications of this work that could be explored or discussed?

3. What are the next steps in this research? How do you plan to investigate further the differential metabolites identified?

4. Discuss any potential limitations or confounding factors that may have influenced the results.

Author Response

In this research, the authors present draft genome sequences for six species from Aspergillus section Terrei and one species from Aspergillus section Nidulantes. The work looks quite interesting, but in its current form there are a lot of questions about it. Therefore, it should not be accepted for publication without serious revision

  1. Discussion is too short. This part should be improved and contained the comparison with the similar articles.

Response: We respectively disagree with the reviewer as results and discussion are combined and make up 8 pages of the manuscript. We have compared the results to all similar studies and feel that expanding it will not provide additional insights.

  1. Are there any broader implications or applications of this work that could be explored or discussed?

Response: We have added some additional text to the conclusion.

  1. What are the next steps in this research? How do you plan to investigate further the differential metabolites identified?

Response: See previous comment.

  1. Discuss any potential limitations or confounding factors that may have influenced the results.

Response: All results have been discussed in detail including such issues. As all experiments were done by experienced people, we do not think it is needed to discuss technical limitations in more detail.

Reviewer 2 Report

The article “Genomic analysis of Aspergillus section Terrei reveals a high potential in secondary metabolite production and plant biomass degradation” by Theobald et al. is devoted to the study of the potential for chemical diversity of the species from Aspergillus section Terrei based on the genome sequences by predicted gene clusters for secondary metabolite production, as well as the potential for plant biomass degradation. In general, the paper provides some basis for the development, utilization and functions for the metabolites from the species of Apergillus section Terrei.

The English is clear, and the references are relevant. I suggest that the section “4. Conclusions” should add some prospects.

Author Response

Response to reviewer 2

The article “Genomic analysis of Aspergillus section Terrei reveals a high potential in secondary metabolite production and plant biomass degradation” by Theobald et al. is devoted to the study of the potential for chemical diversity of the species from Aspergillus section Terrei based on the genome sequences by predicted gene clusters for secondary metabolite production, as well as the potential for plant biomass degradation. In general, the paper provides some basis for the development, utilization and functions for the metabolites from the species of Apergillus section Terrei.

The English is clear, and the references are relevant. I suggest that the section “4. Conclusions” should add some prospects.

Response: Thank you for the positive review. We are glad that you see how the works supports further study of the Terrei section. We have added some prospects to the conclusions section on the evolutionary aspects and biotechnological aspects of the study. We are a bit hesitant to give a high level details on follow-up studies, as it may impact IP protection of current and future work.

Reviewer 3 Report

The study conducted genome mining of several species from Aspergillus section Terrei and identified diverse and abundant potential secondary metabolite gene clusters. By comparing to other Aspergilli, the authors highlighted that this Aspergillus section not only forms a unique phylogenetic cluster, but also exhibits several notable metabolic diversity, such as, the use of a distinctive UV-protection strategy and high potential for plant biomass degradation. This study provides insights in the biotechnology potential of the Aspergillus section Terrei. The manuscript is well-organized overall, but I have a few comments and questions.

Method section 2.6 Generation of SMGC families: Besides backbone and tailoring enzymes, have the authors identified other types of enzymes in the SMGCs? If so, why did the authors only focused on backbone and tailoring enzymes? For the calculation of cluster similarity score, I suggest adding a detailed description of the symbols (p, n, and t) in the equations. Eq.1 and eq. 2 are not in the same format and nxtailoring in eq.3 might be a typo.

How many SMGC families have been identified in section Terrei? In Table S2, I assume the codes in the 'Cluster Family' column correspond to the individual SMGC families, and it appears that there are 167. 

Genetic dereplication analysis: I am also interested in how many SMGC families did not match MIBiG clusters. Were any of those novel SMGC families predominant in section Terrei, but not other species? This could imply a specific evolutionary or ecological adaptation within section Terrei that leads to the production of these novel secondary metabolites. 

Conidia pigmentation analysis: The conidia of A. neoindicus appeared yellowish in Figure 4. Could this discrepancy be related to factors such as printing quality, optical illusions, or perhaps other potential unknown pigments?

Line 192: PpksP might be a typo.

Line 215: Should be Growth profiling.

Table S1: The results for N50 and L50 seems to be exchanged. N50 should represent the length of the sequence at which 50% of the total assembly length is covered, while L50 should represent the number of scaffolds needed to cover 50% of the total assembly length.

Figure 1 B: Second column represents the scaffolds over 2 kb, which is inconsistent with 200 kb mentioned  in Table S1.

Line 243: Similar to the issue noted earlier, N50 should represent the sequence length at which 50% of the total assembly length.

Figure S3: Given the high content of putative cellulases I expected to see growth of strains on cellulose plates. However, for certain species, e.g. A. flavos, A. ellipticus, the images appear distorted. I suggest increasing the resolution or annotating the pictures.

Suggest providing definitions or explanations for abbreviations, e.g., HPLC, upon their first use.

Author Response

The study conducted genome mining of several species from Aspergillus section Terrei and identified diverse and abundant potential secondary metabolite gene clusters. By comparing to other Aspergilli, the authors highlighted that this Aspergillus section not only forms a unique phylogenetic cluster, but also exhibits several notable metabolic diversity, such as, the use of a distinctive UV-protection strategy and high potential for plant biomass degradation. This study provides insights in the biotechnology potential of the Aspergillus section Terrei. The manuscript is well-organized overall, but I have a few comments and questions.

Method section 2.6 Generation of SMGC families: Besides backbone and tailoring enzymes, have the authors identified other types of enzymes in the SMGCs? If so, why did the authors only focused on backbone and tailoring enzymes? For the calculation of cluster similarity score, I suggest adding a detailed description of the symbols (p, n, and t) in the equations. Eq.1 and eq. 2 are not in the same format and nxtailoring in eq.3 might be a typo.

Response: We have edited the format and clarified the formulas. By using backbone and tailoring enzymes to create a cluster score, the algorithm considers all biosynthetic enzymes that are required for the production of secondary metabolites. In our definition, all coding genes assigned to the predicted cluster are assigned to be “tailoring enzymes”, therefore, they cover all predicted enzymatic activities, as mentioned by the reviewer. As an example, the pathway for terreic acid (https://www.uniprot.org/uniprotkb/Q0CJ59/entry) the backbone enzyme atX (a 6-MSA synthase) provides the first step of the terrific acid pathway, followed by the tailoring enzymes atA (decarboxylase), atE (cytochrome P450) and atC (oxidase) to yield the final molecule.

How many SMGC families have been identified in section Terrei? In Table S2, I assume the codes in the 'Cluster Family' column correspond to the individual SMGC families, and it appears that there are 167. 

Response: The numbers in the manuscript were somehow off compared to the Table S2, which may have led to the confusion. The number is indeed 167. We have corrected the numbers in the manuscript throughout. Thank you for catching this.

Genetic dereplication analysis: I am also interested in how many SMGC families did not match MIBiG clusters. Were any of those novel SMGC families predominant in section Terrei, but not other species? This could imply a specific evolutionary or ecological adaptation within section Terrei that leads to the production of these novel secondary metabolites. 

Response: We examined that 17 out of 167 SMGC are known (line 374). The remaining SMGC families were not assigned to a MIBIG cluster in our analysis. We find it difficult to make any hypotheses on evolutionary adaptation, as the MiBiG clusters are derived from experimental determination of cluster size, and these clusters are typically not selected based on evolutionary principles or a specific bioactivity, but more the criteria of the given researcher for finding the cluster. Sometimes, this is a random screen for gene expression or cluster activation by transcription factor overexpression, as in the work of professors Oakley, Wang, Mortensen, or Tang.

Conidia pigmentation analysis: The conidia of A. neoindicus appeared yellowish in Figure 4. Could this discrepancy be related to factors such as printing quality, optical illusions, or perhaps other potential unknown pigments?

Response: This is a good point. Our pictures of fungi across the full Aspergillus genus projects were taken after 10 days of growth, and a number of days of storage at 5 degrees The 300+ species are at quite different stages at this point, which causes some variation in maturity. These were grown on CYA medium, and we believe that the spores are immature at this point. We also have the same isolate on MEAOX plates (Shared to Wikimedia commons: https://upload.wikimedia.org/wikipedia/commons/b/b6/Aspergillus_neoindicus_meaox.png), and there the spores are more developed and completely white. Our auto-color adjustment algorithm also seems to have skewed some of these pictures towards the yellow spectrum.

Line 192: PpksP might be a typo.

Response: PpksP is intended to mean the promoter of the pksP gene, but the text is ambiguous. We have updated the wording.

Line 215: Should be Growth profiling.

Response: Fixed, thank you

Table S1: The results for N50 and L50 seems to be exchanged. N50 should represent the length of the sequence at which 50% of the total assembly length is covered, while L50 should represent the number of scaffolds needed to cover 50% of the total assembly length.

Response: The reviewer misinterpreted this. N is number and L is length.

Figure 1 B: Second column represents the scaffolds over 2 kb, which is inconsistent with 200 kb mentioned in Table S1.

Response: This was an error in Table S1, which is now corrected.

Line 243: Similar to the issue noted earlier, N50 should represent the sequence length at which 50% of the total assembly length.

Response: See response above.

Figure S3: Given the high content of putative cellulases I expected to see growth of strains on cellulose plates. However, for certain species, e.g. A. flavos, A. ellipticus, the images appear distorted. I suggest increasing the resolution or annotating the pictures.

Response Growth on commercial cellulose is poor for most fungi, even those with very good cellulolytic ability (e.g. Trichoderma reesei), likely due to modifications caused by chemical extraction. The images are therefore not distorted, but growth is so poor that is it barely visible on the white cellulose plates.

Suggest providing definitions or explanations for abbreviations, e.g., HPLC, upon their first use.

Response: We have added definitions for all less common abbreviations, but did not include them for common ones, such as DNA, HPLC, etc.